# LIMITATIONS ON ACCURATE, TRUSTED, HUMAN-LEVEL REASONING

## ABSTRACT

Accuracy, trust and human-level reasoning are aspirational goals in artificial intelligence (AI) systems, and there are several informal interpretations of these notions. In this paper, we propose strict, mathematical definitions of accuracy, trust, and human-level reasoning, and demonstrate a fundamental incompatibility between them. We define accuracy of a system as the property that it never makes any false claims when it has the ability to abstain from making a prediction on any input, trust as the assumption that the system is accurate, and human-level reasoning as the property of an AI system always matching or exceeding human capability. Our core finding is that—for our formal definitions of these notions—an accurate and trusted AI system cannot be a human-level reasoning system: for such an accurate, trusted system there are task instances which are easily and provably solvable by a human but not by the system. We note that we consider strict mathematical definitions of accuracy and trust, and it is possible for real-world deployments to instead rely on alternate, practical interpretations of these notions. We show our results for program verification, planning, and graph reachability. Our proofs draw parallels to Gödel's incompleteness theorems and Turing's proof of the undecidability of the halting problem, and can be regarded as interpretations of Gödel's and Turing's results.

## 1 INTRODUCTION

Rapid advancements in artificial intelligence have intensified focus on achieving human-level reasoning across diverse tasks (Morris et al., 2024; Feng et al., 2024). AI systems capable of human-level reasoning have the potential for vast societal benefits through transformative impacts on nearly every aspect of society, including healthcare (Singhal et al., 2025), scientific research (Wang et al., 2023), education (Wang et al., 2024), sustainability (Rolnick et al., 2022), and economic growth (Chui et al., 2023). At the same time, development of such powerful systems necessitates a foundational emphasis on safety and trustworthiness. Consequently, there has been significant interest in ensuring safety and trust for AI systems (Bostrom, 2014; Amodei et al., 2016; Russell, 2019; Jacovi et al., 2021; Tegmark & Omohundro, 2023).

In this work, we point out a fundamental tension between the requirements of an AI system being accurate and trusted, but also matching or exceeding human reasoning capabilities, i.e. being a human-level reasoning system. There are several interpretations of accuracy, trust and human-level reasoning and our result does not preclude achieving these desiderata simultaneously under more relaxed interpretations that could still be useful in many practical applications. Therefore, to understand the limitations pointed out by our result, it is important to first understand our formalizations of these notions, and we will immediately proceed with defining these notions. We start by first defining an AI system for a given task.

**Definition 1.1** (AI system). *We define an AI system as a system which takes an instance of a task, and either solves the instance or abstains from giving an answer for the instance (for instance by outputting 'don't know'). We allow the AI system to be randomized, for example it could abstain with some probability (with respect to its internal randomness) on an instance, and output a solution otherwise.*

Definition 1.1 simply formalizes the notion of a system which solves instances of a task. In this paper, we will consider the tasks of program verification, planning and determining graph reachability

(defined rigorously later). Note that we allow the system to abstain from providing an answer for some instance if it so determines, which could be important from the perspective of safety (Geifman & El-Yaniv, 2017). Now, we define the notion of accuracy.

**Definition 1.2** (Accuracy). *We define a system to be accurate if it does not make any false claims, i.e., for every instance it either answers the instance correctly or abstains from answering it.*

As an example, in the context of verifying that a program has some specified property (such as always terminating), the system is accurate if it does not classify a program as having the desired property if it does not have that particular property. Our definition allows the system to abstain from answering an instance if it is uncertain, but it requires the system to be correct whenever it outputs an answer. Note that our notion of accuracy is closely related to notions of factuality, reliability and, in some situations, even safety. Often abstention is preferred over a confident but incorrect response as the consequences of an incorrect response may be severe. Even small probabilities of error may not be tolerable for mission-critical tasks, especially as system capabilities grow (Amodei et al., 2016; Tegmark & Omohundro, 2023). Next, we define trust as simply the assumption of accuracy.

**Definition 1.3** (Trust). *We define trust to be the assumption that the system is accurate.*

To elaborate on the definition, if a system is trusted then it is assumed that the system is accurate. As a remark, we note that our results are agnostic to whether trust in the system stems from theoretical proofs, empirical verification, or some combination of these, we only require that when deploying the system there is an assumption that it is accurate. We also note that accuracy does not necessarily imply trust, or vice versa. Accuracy is an underlying property of the system being consistent and not making false claims. It is possible that some analysis of the system cannot identify this property or is incorrect, leading to a lack of trust or mistaken trust. For example, a system could actually be accurate but not trusted because existing empirical or theoretical tools are insufficient to establish accuracy. Similarly, a system could actually be inaccurate but still trusted by users, such as when the trust rests on empirical evidence which is incomplete, or on incorrect theoretical assumptions.

Finally, we need to formally define a human-level reasoning system in order to mathematically investigate its limitations.

**Definition 1.4** (Human-level reasoning). *We define a system to be a human-level reasoning system if for every task instance such that a human has a provably correct solution for that instance, the system can also solve the instance with some non-zero probability. Similarly, the system is not a human-level reasoning system if there exists some task instance which can be easily and provably solved by a human, but the system can never solve the instance (for probabilistic systems, the probability of the system solving the instance is 0).*

Our definition draws on the common view that a human-level reasoning system for a task such as program verification should be at least as capable as a human on that task. In particular, if there are explicit task instances which can be provably solved by humans (for example, explicit programs which the humans can easily and provably certify as having the desired property) but cannot be solved by the system, then the system is not a human-level reasoning system as per our definition.

Our definition draws on some similar notions of artificial general intelligence (AGI). It is well-accepted that there is no well-accepted definition of AGI—or even of intelligence itself (Legg & Hutter, 2007; Legg et al., 2007)—but the term AGI is usually understood to mean that the AI system should be at least as capable as humans across diverse tasks. The term *superintelligence* is also used in the context of AGI (Bostrom, 2014; Morris et al., 2024). For example Bostrom (2014) defines superintelligence as *"any intellect that greatly exceeds the cognitive performance of humans in virtually all domains of interest"*, and Morris et al. (2024) defines Level 5 AGI, which they term artificial superintelligence, as *"outperforming 100% of humans"* on a *"wide range of non-physical tasks"*. One distinction between these notions of superintelligence and our definition of human-level reasoning is that our definition does not require the AI system to necessarily *outperform* humans, but it does require the system to do at least as well as humans on all task instances.

In our definition, when we say that a human has a provably correct solution, we mean that the human can provide a scientifically acceptable proof. In this paper whenever we make claims about humans being able to solve problems, we provide such proofs. To probe this point and Definition 1.4 further, we consider an analogy to chess — a domain for which we have had advanced AI systems for quite some time. Consider a future proposed human-level reasoning system, which is proficient at chess

among other things. If there were explicit chess positions which most human chess players can solve provably without too much difficulty, but the proposed human-level reasoning system struggled on those positions, then the proposed system does not capture some aspects of human cognition, and hence is arguably not actually a human-level reasoning system.[1] Similarly, in our paper we will demonstrate explicit instances of certain tasks for which we provide solutions with short, scientifically acceptable proofs which are also rather simple, but these instances cannot be solved by AI systems having certain properties.

We now state our main result, that it is not possible for a human-level reasoning system to be both accurate and trusted, as per our definitions of accuracy, trust and human-level reasoning. In other words, the notions of accuracy, trust and human-level reasoning are mutually incompatible — any system can have at most two of these three properties.

**Theorem 1.5.** *If an AI system is accurate and trusted, then it cannot be a human-level reasoning system. In particular, it is not a human-level reasoning system for the tasks of program verification, planning and determining graph reachability.*

Theorem 1.5 points out a fundamental limitation of a human-level reasoning system: such a system cannot be both accurate and trusted. Similarly, if there is some trusted AI system, then either that system is not actually accurate, or it is not a human-level reasoning system. We prove this result in Section 3. While much of our proof technique mimics Gödel's proof of his incompleteness theorems (Gödel, 1931) (and also Turing's proof of the undecidability of the halting problem (Turing, 1937)), the argument we make is not in the context of axiomatic system and theorem proving but in the context of an AI system that needs to solve certain task instances of applications such as program verification or planning. Our proofs are self-contained in this context and do not require knowledge of formal axiomatic reasoning or logical rules of deduction. Thus rather than viewing the results as limitations of systems of logic, they should be viewed as limitations of AI systems.

We also consider a relaxation of accuracy which requires the AI system to be calibrated with respect to its predictions, as opposed to Definition 1.2 which requires the system to be always correct unless it abstains. We refer to this notion calibration, and for the case of program verification calibration requires that if the AI system outputs that some program terminates with some probability $p$, then that program should actually terminate with probability approximately $p$. In Section 4 we show a similar limitation as in Theorem 1.5 for AI systems which are calibrated.

## 2 RELATED WORK

In this section, we discuss some more related work on human-level reasoning, accuracy and trust in AI, and limitations of AI in the context of Gödel's results.

**Human-level Reasoning Systems.** Though not termed as "Artificial General Intelligence (AGI)" until more recently (Goertzel & Pennachin, 2007), the concept of machines which match or surpass the cognitive capabilities of humans dates back to the earliest days of AI (Turing, 1950; McCarthy et al., 1955; Minsky, 1961). Due to recent advances in foundation models such as large language models (Bommasani et al., 2021), there has been significant interest and capital investments in developing systems capable of human-level reasoning both from the private sector and from governments (Maslej et al., 2025).

**Accuracy, Reliability and Trust in AI.** Concerns around risks associated with advanced AI systems similarly date back to early days of AI (Turing, 1951; Wiener, 1950). With growing system capabilities, there has been significant recent focus on ensuring safety and trust in the context of AI systems (Future of Life Institute, 2024; for AI Safety, 2025). We refer the interested reader to several recent surveys and roadmaps for ensuring safety and trust in highly-capable, general purpose AI systems (Bengio et al., 2024; Chua et al., 2024; Chen et al., 2024; Bengio et al., 2025). It is also

---

[1] We note that many current advanced chess engines still struggle to evaluate certain positions which are relatively easy for human experts (Doggers, 2017; Zahavy et al., 2023). However, this is likely a result of the these engines being 'narrow' in terms of their approach and reasoning, and we believe that a proposed human-level reasoning or superintelligent system which is purported to excel on chess should have the ability to solve such instances.

important to recognize that AI safety and trust encompass many facets beyond those considered in our definitions. For example, even formally specifying safety objectives can be challenging for complex tasks (Amodei et al., 2016), which introduces additional challenges to develop safe AI systems beyond those pointed out in our work.

**Gödel and Turing's results.** Fundamental limits on theorem proving and program verification were famously established by Gödel's incompleteness theorems and Turing's undecidability results. Gödel showed that in any sufficiently expressive formal system, there exist true statements (also called Gödel statements) that cannot be formally proven within the system (Gödel, 1931). Building on this, Turing proved that the Halting Problem—determining whether an arbitrary program halts on a given input—is undecidable (Turing, 1937), meaning no algorithm can solve it for all possible programs. These results imply that fully automatic verification of arbitrary program behavior, such as ensuring termination, is provably impossible in the general case. Our result uses similar ideas to draw a separation between the abilities of a safe, trusted AI system and humans.

**Penrose-Lucas argument, and implications of Gödel's results for AI.** Several arguments have been made for why Gödel's result imply that AI can never match humans, the most famous of which are perhaps due to Penrose (Penrose & Gardner, 1989) and Lucas (Lucas, 1961). To summarize very briefly, Penrose and Lucas have argued that incompleteness does not apply to humans since they can see the truth of Gödel statements, and therefore humans can have mathematical insights that Turing machines cannot (Wikipedia). This argument is quite contested, and several objections have been raised against it (Chalmers, 1995; LaForte et al., 1998; Kerber, 2005) — again going back to Turing (Turing, 1950) — with a core objection being that humans also cannot be certain that their own reasoning process is sound.

The goal of our work is distinct from that of Penrose and Lucas, and we do not aim to show a separation between *any* AI system and human reasoning. Instead, we prove a more restricted but rigorous result: that *accurate, trusted* AI systems (under formal definitions of those terms) are necessarily unable to solve certain problems that humans can solve with provable correctness. The assumption of accuracy and trust is crucial (as will be evident from our proofs) — it allows humans to conclude the correctness of some outputs even when the AI system, by its own constraints, must abstain.

We also note that there are some other limitations of AI which have been pointed out by using Gödel and Turing's results, such as the impossibility of "containing" superintelligence (Alfonseca et al., 2021), and the necessity of hallucinations in a certain formal model (Xu et al., 2024), see the survey Brcic & Yampolskiy (2023) for other results similar to these.

## 3 TECHNICAL RESULTS

In this section, we discuss our main technical results regarding limitations of accurate, trusted, human-level reasoning for program verification, planning, and graph reachability.

### 3.1 PROGRAM VERIFICATION

The first task we consider is program verification, more specifically the task of determining if a given program always halts. Program verification (also formal verification) is a foundational problem in computer science and software engineering, with critical implications for ensuring the reliability, safety, and correctness of software systems (Hoare, 1969; Clarke et al., 2018).

**Definition 3.1** (Program verification). *We define a program to be* well-behaved *if it terminates on every input (for randomized programs, the program terminates with probability 1). In the program verification problem, the system is given a program instance and it classifies the instance as being 'well-behaved', 'not well-behaved' or abstains from making a prediction (outputs 'don't know'). Accuracy for program verification requires that the system never outputs that a well-behaved program is not well-behaved, and vice versa. The system is trusted if we assume that the system is accurate. Note that the system is not a human-level reasoning system if there is a well-behaved program which can be easily proven to be well-behaved by a human, but for which the program always abstains from making a prediction.*

Claim (informal): If $A$ is accurate then Gödel_program is well-behaved, but $A$ cannot output that Gödel_program is well-behaved.

```
procedure Gödel_program
    if A(Gödel_program) == 'well-behaved'
then
        while true do
        end while
    else
        return 0
    end if
end procedure
```

**Proof sketch:**

- If $A$ outputs that Gödel_program is well-behaved, then the program enters an infinite loop.
- If $A$ is accurate, this is a contradiction, hence $A$ cannot output Gödel_program is well-behaved.
- If $A$ does not output Gödel_program is well-behaved, then the program immediately terminates and hence is well-behaved.

Figure 1: Sketch of the basic argument for program verification, for the case when the AI system $A$ is well-behaved (i.e., always terminates) and deterministic. If $A$ is accurate, it cannot determine if Gödel_program is well-behaved. However, since the condition of $A$ being accurate is satisfied for a trusted system, it is possible to prove that Gödel_program is well-behaved for a trusted system. Therefore, a accurate, trusted system cannot solve this instance, even though it is provably solvable.

Our definition of program verification (Definition 3.1) and our results are for the property of the program halting. In Appendix B, we extend the result to a general class of properties of programs, including those relevant from the perspective of safety.

**Theorem 3.2.** *If a system is accurate and trusted, then it cannot be a human-level reasoning system for program verification.*

*Proof.* Our proof can be regarded as a restatement of Gödel's proof, presented here in the context of program verification. In Fig. 1 we sketch the basic version of the argument, for the case when the AI system $A$ is deterministic and well-behaved.

We now proceed with the proof, which relaxes the assumptions in Fig. 1 of $A$ being deterministic and well-behaved. Consider any AI system $A$ which takes as input program $P$ and outputs 'well-behaved', 'not well-behaved' or 'don't know'. Our construction will leverage the *trace* of the system $A$ run on some program $P$, which is just the execution trace of the system $A$ when it is given program $P$ as input. Now consider the program instance in Algorithm 1.

---

**Algorithm 1** Gödel_program

---

1: **procedure** Gödel_program$(P, T)$
2:     **if** $T$ is not a syntactically-valid trace of the system $A$ evaluated on program $P$ **then**
3:         **return** 0
4:     **else if** $T$ outputs $P$ is 'well-behaved' **then**
5:         **if** $(P, T)$ is a valid input to program $P$ **then**
6:             **return** concatenation('Not ', $P(P, T)$)
7:         **end if**
8:     **else**
9:         **return** 0
10:     **end if**
11: **end procedure**

---

Note that Gödel_program involves running the program $P$ on the input pair $(P, T)$. Gödel_program checks that $(P, T)$ is a valid input type to the program $P$, and we can also regard the input $(P, T)$ as one input to $P$ that is a pair of entities: a program $P$ and a trace $T$. We now show that if $A$ is accurate, then Gödel_program is well-behaved.

**Lemma 3.3.** *If A is accurate, then* Gödel_program *is well-behaved.*

*Proof.* We first claim that the **if** and **else if** conditions in steps 2, 4 and 5 always terminate (if the program enters those steps). Step 2 always terminates since it involves checking if every step of the trace $T$ is a valid step which the AI system $A$ can take. Step 4 just involves checking the output of the trace, and step 5 also terminates since the step involves checking if the input matches the required format for the program.

Now consider the case when the **else if** condition in step 4 is satisfied. This happens when the trace $T$ concludes that $P$ is well-behaved, and since the system $A$ is accurate, then $P$ must be well-behaved in that case. Hence the execution in step 6 always terminates, and hence Gödel_program terminates for that input.

On the other hand, if the trace does not conclude that $P$ is well-behaved, then the program enters the **else** condition in lines 8 and 9, and immediately terminates. Therefore if $A$ is accurate, then Gödel_program is well-behaved. □

We note that that the assumption of $A$ being accurate is satisfied for a trusted system $A$. Therefore, if a system is accurate and trusted, then a human can prove that Gödel_program well-behaved, indeed the proof of Lemma 3.3 provides a short proof of this. Next, we show that the system $A$ cannot solve this instance.

**Lemma 3.4.** *A can never output 'well-behaved' for* Gödel_program.

*Proof.* The proof is by contradiction. We first note that Gödel_program is a deterministic program, and can only have a single output for a given input. Suppose there is a valid trace $T_G$ for the AI system $A$ which outputs 'well-behaved' for Gödel_program. Consider Gödel_program(Gödel_program, $T_G$). Then the output of step 6 differs from Gödel_program(Gödel_program, $T_G$), which is also the output of Gödel_program on the input (Gödel_program, $T_G$). This is a contradiction since a deterministic program cannot have two outputs on the same input, and hence $A$ can never output 'well-behaved' for Gödel_program. □

Therefore, if $A$ is accurate and trusted, then it cannot be a human-level reasoning system. □

## 3.2 PLANNING

The next task we consider is planning, a long-studied task in artificial intelligence (LaValle, 2006; Russell & Norvig, 2016). Planning is also considered important for general-purpose cognitive capabilities (Goertzel & Pennachin, 2007).

**Definition 3.5** (Planning). *In a planning problem we are given a sequence of states, a set of associated moves, a start state and a desired goal state. For any state* u *and move pair, there is a an explicit program (which is provided as part of the problem specification) which returns the next state (certain moves may be illegal and may return in 'not allowed' states). The task is to find a sequence of moves which end up in the goal state from the start state, or to prove that it is not possible to reach the goal state from the start state.*

For clarity of exposition, we consider deterministic planning instances and deterministic AI systems in this section. In Appendix A.1, we also consider randomized AI systems and randomized problem instances.

**Theorem 3.6.** *If a deterministic AI system is accurate and trusted, then it cannot be a human-level reasoning system for planning. In particular, for such a system there is a planning problem instance for which the system outputs 'don't know' but there is a short proof that the planning problem has no winning moves.*

We prove this by reduction from a variant of program verification that involves checking whether a given program, input pair halts.

### 3.2.1 Halting for a specific program input instance

We first define the problem of checking halting for a specific program, input instance.

**Definition 3.7** (Halting for a specific program input instance). *Given a deterministic program and an input for the program, check whether the given program halts or does not halt on the given input.*

We show the following result for this halting problem.

**Theorem 3.8.** *If a deterministic system is accurate and trusted, then it cannot be a human-level reasoning system for the task of determining whether a program halts on a specific input instance.*

We note that Theorem 3.6 follows from Theorem 3.8. This is because we can reduce the halting problem for a program-input pair to a planning instance. Given a program $P$ and input $I$, we construct a planning problem where the states correspond to the configurations of $P$ during its execution on $I$, and the moves represent single-step transitions between these configurations. The start state is the initial configuration of $P$ on input $I$, and the goal state is a halting configuration of $P$. The planning task is to determine whether a sequence of moves exists that leads from the start state to the goal state—this is equivalent to determining whether $P$ halts on $I$. Hence, if a safe and trusted system could solve all such planning instances, it would be able to solve the halting problem in Definition 3.7, contradicting Theorem 3.8. This establishes that, under our definitions, a safe and trusted system cannot be a human-level reasoning system for planning.

The proof of Theorem 3.8 appears in Appendix **??**, and is similar to the proof of Theorem 3.2, and also the next result, Theorem 3.12. In Appendix A.1, we prove a similar version of Theorem 3.8 where the program provably halts on the input, but the AI system $A$ cannot determine so. This proof requires an additional assumption that $A$ is also well-behaved, i.e. it always terminates on an input, which can be ensured by having a fixed time limit on the execution of $A$. Note that since determining halting on a specific program input instance reduces to planning, this shows that for a safe, trusted and well-behaved system there are planning instances where humans can provably find a feasible plan, but the system will not be able to solve the instance.

## 3.3 Graph reachability

We now consider the graph reachability problem. Graph reachability can also be regarded as an instance of the search problem, another fundamental problem in artificial intelligence with numerous applications (Russell & Norvig, 2016). Graph reachability is closely connected to the planning problem that we defined in the previous section, a distinction we make is that for planning problems the state space can be potentially infinite, whereas for reachability we consider finite-sized graphs.

**Definition 3.9** (Graph reachability). *Given a (possibly directed) graph $G$ and a source-sink pair $u, v$, check whether $v$ is reachable from $u$. We allow the graph to be defined via an explicit program (which is provided as part of the problem specification). The program takes any vertex $v$ and returns the adjacency list of $v$.*

We show that accurate, trusted AI systems need time almost as large as the size of the considered graph to solve certain reachability instances which actually admit a simple, solution. As in Section 3.2, we consider deterministic AI systems in this section for ease of exposition. In Appendix A.2, we extend to randomized AI systems.

**Theorem 3.10.** *For any $T > 0$, a fixed constant $c$, and any accurate, trusted, deterministic AI system, there is a graph reachability problem instance of size $T$, for which the accurate, trusted, deterministic AI system outputs 'don't know' if it is run for time at most $T - c$, but there is a short, constant-sized proof that the answer is 'not reachable'.*

We prove this by reduction from a variant of program verification that involves checking whether a given program halts within a fixed amount of time.

### 3.3.1 Time-bounded halting

**Definition 3.11** (Time-bounded halting). *Given a deterministic program and an input for the program, check whether the given program halts or does not halt on the given input in a given number of time steps $T$.*

**Theorem 3.12.** *If a deterministic system $A$ is accurate and trusted, then it cannot be a human-level reasoning system for time-bounded halting. Specifically for a deterministic, accurate, trusted system $A$ and for any $T > 0$ and a fixed constant $c$, there is a program for which there is a short, constant-sized proof that it does not halt in $T$ steps, but $A$ will output 'don't know' if it runs for time at most $T - c$.*

We note that Theorem 3.10 follows from Theorem 3.12. This is because time-bounded halting can be reduced to graph reachability (similar to the reduction for planning), where the graph is defined by the states of the program and the goal is to determine if the program reaches a halting state. Theorem 3.12 shows that there is a graph of size $T$ where the AI system needs time nearly $T$ to solve reachability, but a human can prove a constant sized proof that the sink vertex is not reachable from the source.

We now prove Theorem 3.12.

*Proof of Theorem 3.12.* Consider any deterministic AI system $A$ which takes as input program $P$, input $I$, and time limit $T$ and outputs 'halts in given time limit', 'does not halt in given time limit' or 'don't know'. Consider the program in Algorithm 2, defined for some fixed time limit $T > 0$.

---

**Algorithm 2** Turing_T

---

```
1: procedure Turing_T(Program P, Input I)
2:     if A(P, I, T) == 'does not halt in given time limit' then
3:         return 0
4:     else
5:         while true do                                    ▷ run indefinitely
6:         end while
7:     end if
8: end procedure
```

---

We define
$$\texttt{self\_Turing\_T}(P) = \texttt{Turing\_T}(P, P).$$

Now consider `self_Turing_T(self_Turing_T)`.

**Lemma 3.13.** *If $A$ is accurate, then* `self_Turing_T(self_Turing_T)` *does not halt in time $T$. Moreover, for a fixed constant $c$, if $A$ is accurate and is run for time at most $T - c$ then $A$ will output 'don't know' on whether* `self_Turing_T(self_Turing_T)` *halts in time at most $T$.*

*Proof.* If `self_Turing_T(self_Turing_T)` halts, it can only be because it enters the **if** block in line 3. However, it only enters this block if $A$ determines that it does not halt in time $T$. Since $A$ is accurate, if the program enters the **if** block in line 3 then it must not halt in time $T$, and hence `self_Turing_T(self_Turing_T)` cannot halt in time $T$.

Note that the execution of steps 2 and 3 of the program only take some fixed constant $c$ steps outside the execution of $A$ on (`self_Turing_T, self_Turing_T`, $T$). Therefore if the AI system $A$ runs for time $T'$ and outputs that `self_Turing_T(self_Turing_T)` does not halt in time $T$, then `self_Turing_T(self_Turing_T)` halts in time $T' + c$. If $T' < T - c$, then the program does halt in total time $T$, which contradicts accuracy. Therefore, a accurate AI system $A$ must output 'don't know' if it runs for time at most $T - c$, for some fixed constant $c$. $\square$

As in the previous proofs, note that the assumption of $A$ being accurate is satisfied for a trusted system $A$, therefore for a trusted system we have a short proof that `self_Turing_T(self_Turing_T)` does not halt in time $T$, even though $A$ cannot solve this instance if it is accurate and run for time at most $T - c$. $\square$

At the end of Section 3.2.1, we discussed an additional result about planning in the case where a feasible plan exists. We also show a similar result for graph reachability. In Appendix A.2, we prove a similar version of Theorem 3.12 under an additional assumption that $A$ always terminates in time $T$. We show that for such a system $A$ there is an instance which provably halts in time $T + c$ (for some constant $c$) if $A$ is accurate, but the accurate system $A$ cannot determine so. As before, since

determining halting within a fixed time limit reduces to graph reachability on finite-sized graphs, this shows that for a accurate, trusted system with an upper bound on its running time, there are graph reachability instances where humans can provably find a path, but the system will not be able to solve the instance in time slightly less than the size of the graph.

## 4 Impossibility Result for Calibration

In this section, we define a relaxed notion of accuracy, which we term as calibration. This notion is derived from the usual notion of calibration, a well-studied notion for ensuring reliability of a model's prediction (Dawid, 1982; Van Calster et al., 2019).

**Definition 4.1** (Calibration for Program Verification). *We define a system $A$ to be calibration-safe if for any program $P$ with input $I$:*

1. *If $A$ outputs 'halts' with some probability $p > 0$ when given program $P$ and input $I$, then the probability of $P$ halting on $I$ lies in $[p - 0.25, p + 0.25]$.*

2. *If $A$ outputs 'does not halt' with some probability $p > 0$ when given program $P$ and input $I$, then the probability of $P$ not halting on $I$ lies in $[p - 0.25, p + 0.25]$.*

Note that similar to the definition of accuracy (Definition 1.2), calibration does not put any requirement on the system if it decides to abstain with probability 1 on a given input. We show that if a system is calibration-safe, and in addition is also well-behaved, then it fails on certain instances which provably terminate with good probability.

**Theorem 4.2.** *If the AI system $A$ is well-behaved and calibration-safe for program verification, then there is a program $P$ which provably halts with probability at least $0.99$, but $A$ abstains with probability 1 on the program $P$.*

We note that Theorem 4.2 implies a similar impossibility result as Theorem 3.2 but with a relaxed notion of accuracy and a corresponding notion of trust. In the context of calibration, trust in Definition 1.3 is the assumption that the system is calibrated. Then Theorem 4.2 implies that for a well-behaved, calibrated and trusted system $A$, there is a program which can be proven to halt with probability at least 0.99, but $A$ will abstain with probability 1 on the program. Therefore, a well-behaved, calibrated and trusted system cannot be a human-level reasoning system.

Theorem 4.2 is proved in Appendix A.3. The proof is similar to earlier proofs, but requires an extra step of using a best arm identification algorithm from the multi-armed bandit literature to determine if the probability of the system giving a certain answer is greater than some threshold.

## 5 Discussion

Our results show that accuracy, trust and human-level reasoning are mutually incompatible. We further discuss implications of the result and some possible critiques and clarifications.

- *Circumventing the results by augmenting the AI system:* One may attempt to circumvent the impossibility result by augmenting the AI system, such as by appending new axioms to the system if it is formalized axiomatically. For instance, one could solve Gödel_program (Algorithm 1) defined with respect to some AI system $A$, by designing a new iteration of $A$, say $A'$, which is trained to solve the Gödel_program instance for $A$. However, since our construction is inherently self-referential, this strategy only pushes the problem one step further. For any such extension $A'$, we can construct a new version of Gödel_program defined with respect to $A'$, and the same impossibility result applies again.

- *Worst-case nature of the results:* While we demonstrate specific task instances which are not solvable by certain systems, we note that the system could still solve a vast number of interesting instances. However, the result still points to certain barriers which cannot be overcome by accurate, trusted systems. Given the significant interest and economic capital being devoted to building accurate or reliable human-level reasoning, we believe it is important to understand the barriers fundamental to any such technology. By way of analogy, Gödel's and Turing's results pointed to fundamental barriers to mathematics and

computation. Though these barriers were worst-case, they identified the limits of what is possible and what is possible. Similarly, we believe that it is important to outline the limits of what is possible in the context of AI and requirements of accuracy and trust. Somewhat more speculatively, note that our constructions rely on self-referential calls to the AI system, and when systems have general-purpose capabilities, such calls may not be implausible.

- *Limitations of human reasoning:* We note that there is a long line of work on studying the limitations of human reasoning in cognitive science and other fields, and it has long been emphasized that human reasoning is resource-bounded and error-prone (Simon, 1957; Tversky & Kahneman, 1974). However, our goal is not to argue for strict superiority of human reasoning over AI, but to show a separation: for accurate, trusted AI systems there are instances that humans can solve, but which are not solvable by the system.

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

# A    ADDITIONAL RESULTS

This section proves some additional results discussed in the main text.

## A.1    IMPOSSIBILITY OF SOLVING PLANNING WHEN A FEASIBLE PLAN EXISTS

In this section, we prove a similar result to Theorem 3.8, for the case where the program terminates on the given input. We also strengthen the result in Theorem 3.8 to allow for randomized AI systems, and randomized programs which may halt with some probability on an input. We first extend Definition 3.7 to allow for randomized programs.

**Definition A.1** (Halting for a specific program input instance for randomized programs). *Given a program, input pair, check whether on the given input the given (possibly randomized) program 'always halts', 'halts on some randomness but not all randomness' or 'never halts'.*

**Theorem A.2.** *If the AI system $A$ is accurate and well-behaved for determining halting on a specific program input pair, then there is a program input instance pair for which there is a short proof that the instance always terminates, but $A$ cannot determine that the instance always terminates (the probability of $A$ giving the answer 'always halts' is 0).*

Note that the condition 'If the AI system $A$ is accurate' is satisfied when $A$ is trusted, and hence for a accurate, trusted, well-behaved AI system there are program, input instances which the system cannot solve, but for which there is a short proof that the instance terminates. We now prove Theorem A.2.

*Proof of Theorem A.2.* Consider Algorithm 3.

---
**Algorithm 3** Turing_program_v2

---
1: **procedure** Turing_program_v2(Program $P$, Input $I$)
2:    **if** $A(P, I)$ == 'always halts' **then**
3:        **while** true **do**                                    ▷ run indefinitely
4:        **end while**
5:    **else**
6:        **return** 0
7:    **end if**
8: **end procedure**

---

Now define
$$\texttt{self\_Turing\_program\_v2}(P) = \texttt{Turing\_program\_v2}(P, P).$$
We consider:
$$\texttt{self\_Turing\_program\_v2}(\texttt{self\_Turing\_program\_v2})$$

**Lemma A.3.** *If $A$ is accurate and well-behaved then* self_Turing_program_v2(self_Turing_program_v2) *always halts, but the accurate AI system $A$ cannot determine that it always halts.*

*Proof.* Note that the **if** condition in step 2 always terminates if $A$ is well-behaved. Suppose $A$ outputs 'always halts' on some randomness. Then, self_Turing_program_v2(self_Turing_program_v2) does not halt on some randomness. If $A$ is accurate, then this is a contradiction. Therefore, if $A$ is accurate then it must output 'always halts' with 0 probability.

Note that if $A$ does not output that self_Turing_program_v2(self_Turing_program_v2) 'always halts', then the program enters the **else** condition and immediately terminates, and therefore halts. Therefore if $A$ is accurate and well-behaved, then self_Turing_program_v2(self_Turing_program_v2) always halts.

□

□

## A.2 Impossibility of solving feasibility when a path exists in the graph

We prove a similar result to Theorem 3.12 in this section, for the case where the program terminates on the given input within some time bound. As in Appendix A.1, we also strengthen the result to allow randomized AI systems. We first extend Definition 3.11 to allow for randomized programs.

**Definition A.4** (Time-bounded halting for randomized programs). *Given a program, input pair and a time limit $T$ on the number of execution steps, check whether on the given input the given (possibly randomized) program 'always halts in given time limit', 'halts in given time limit $T$ on some randomness but not all randomness' or 'never halts in given time limit'.*

**Theorem A.5.** *If an AI system $A$ is accurate and always halts in some time $T$, then for a fixed constant $c$ and the time limit $T + c$, there is a program, input pair for which there is a short, constant-sized proof that the instance always halts in at most $T + c$ steps, but for a accurate AI system $A$ which always halts in time $T$ the probability of $A$ giving the answer 'always halts in given time limit' is 0.*

*Proof of Theorem A.5.* Consider Algorithm 4, where $T$ is the upper bound on the running time of the AI system $A$, and $c$ is some fixed constant which is the running time of executing step 2 after $A$ terminates and the **if** condition in step 2 is not satisfied, and then executing steps 5 and 6. Therefore, $T + c$ is an upper bound of the running time of the program when it enters the **else** condition in line 5.

---

**Algorithm 4** Turing_T_v2

1: **procedure** Turing_T_v2(Program $P$, Input $I$)
2:     **if** $A(P, I, T + c) ==$ 'always halts in given time limit' **then**
3:         **while** true **do**                                        ▷ run indefinitely
4:         **end while**
5:     **else**
6:         **return** 0
7:     **end if**
8: **end procedure**

---

We define

$$\texttt{self\_Turing\_T\_v2}(P) = \texttt{Turing\_T\_v2}(P, P)$$

Consider:

$$\texttt{self\_Turing\_T\_v2}(\texttt{self\_Turing\_T\_v2})$$

**Lemma A.6.** *If $A$ is accurate and always terminates in time $T$, then* self_Turing_T_v2(self_Turing_T_v2) *always halts in time at most $T + c$. Moreover, if $A$ is accurate then it has 0 probability of giving the answer 'always halts in given time limit' on whether* self_Turing_T_v2(self_Turing_T_v2) *halts in time at most $T + c$.*

*Proof.* Note that by the definition of $c$, the execution of steps 2, 5 and 6 of the program only take $c$ steps outside the execution of $A$ on the input (self_Turing_T_v2, self_Turing_T_v2, $T + c$).

Suppose $A$ outputs 'always halts in given time limit' on the given input on some randomness. Whenever $A$ outputs 'always halts in given time limit', the program enters an infinite loop and never halts. This is a contradiction if $A$ is accurate, and hence if $A$ is accurate it outputs 'always halts in given time limit' with probability 0.

Now, if $A$ does not output 'always halts in given time limit' on the input, then the program will enter the **else** block and immediately halt. Since $A$ runs for at most $T$ steps, the program then halts in time at most $T + c$. Therefore, if $A$ is accurate then the program always halts in time at most $T + c$.

$\square$

$\square$

## A.3 PROOF OF THEOREM 4.2

**Theorem 4.2.** *If the AI system $A$ is well-behaved and calibrated for program verification, then there is a program $P$ which provably halts with probability at least 0.99, but $A$ abstains with probability 1 on the program $P$.*

*Proof of Theorem 4.2.* Consider Algorithm 5. Throughout the proof we assume $A$ is well-behaved, i.e. it always terminates. Our construction involves a program which does not take any input, i.e. $I = \phi$. The program involves identifying whether the probability $p$ of A outputting 'halts' when given Gödel_program_random as input is greater than 0.5 or not. We use a simple best arm identification procedure for this, for example the algorithm of Karnin et al. (2013).

---

**Algorithm 5** Gödel-program_random

1: **procedure** Gödel_program_random
2:     Let arm$_1$ have the distribution Bernoulli(0.5)
3:     Let arm$_2$ correspond to running $A$ with Gödel_program_random as input, with the result of the arm pull being 1 if $A$(Gödel_program_random) == 'halts', and 0 otherwise.
4:     Run Best-Arm-Identification algorithm from Karnin et al. (2013, Algorithm 1) with confidence parameter $\delta = 0.01$ to determine whether arm$_2$ is better than arm$_1$
5:     **if** arm$_2$ is better than arm$_1$ **then**
6:         **while** true **do**                                    ▷ run indefinitely
7:         **end while**
8:     **else**
9:         return 0
10:    **end if**
11: **end procedure**

---

Note that for any $\epsilon > 0$, if $p = 0.5 + \epsilon$ then arm$_2$ is better than arm$_1$, otherwise if $p = 0.5 - \epsilon$ then arm$_1$ is better than arm$_2$. While we can use any suitable multi-armed bandit algorithm in our construction, here we use Karnin et al. (2013), which has the guarantee that if it is provided with two arms with a gap of $\epsilon$, then it finds the better arm with probability $1 - \delta$ using $O\left(\frac{1}{\epsilon^2} \log\left(\frac{1}{\delta} \log\left(\frac{1}{\epsilon}\right)\right)\right)$ arm pulls. This bound is known to be optimal (Jamieson et al., 2014), though in our case since we do not care about the optimal rate we could have also used earlier sub-optimal procedures (Even-Dar et al., 2002). We also note that if $\epsilon = 0$, then the best arm identification procedure will terminate with probability at most $10\delta$. Though we have not seen the case of $\epsilon = 0$ being directly covered by the guarantees of best arm identification procedures, this claim for $\epsilon = 0$ follows from a simple argument which treats the best arm identification procedure as a black-box. To verify, note that the sequence of observations up to $t$ steps is $\delta$-close in TV distance for any $p \in [0.5 \pm 1/\text{poly}(t, \delta)]$ (where $\text{poly}(t, \delta)$ is some polynomial of $t$ and $\delta$). Therefore for $\epsilon = 0$ and any finite $t$, if the best arm identification procedure terminates in $t$ steps with probability more than $10\delta$, then it will have a failure probability more than $\delta$ for some $p \in [0.5 \pm 1/\text{poly}(t, \delta)]$—which is a contradiction with the guarantee of the procedure. Therefore, for $\epsilon = 0$ the best arm identification procedure terminates with probability at most $10\delta$.

We are now ready to prove the result.

**Lemma A.7.** *If $A$ is calibrated and well-behaved then* Gödel_program_random *halts with probability at least 0.99, but the calibrated AI system $A$ will output 'don't know' with probability 1 on* Gödel_program_random.

*Proof.* We consider three cases.

1. $p \in (0.5, 1]$: Note that in this case with probability at least 0.99 the best arm identification procedure determines that arm$_2$ is better than arm$_1$. Therefore, the program goes into the infinite **while** loop and never terminates with probability at least 0.99. In this case, $A$ is not calibration safe, since it claims that the program terminates with probability $p > 0.5$.

2. $p = 0.5$: As argued above, in this case the best arm identification procedure terminates with probability at most $10\delta = 0.1$. Therefore, Gödel_program_random terminates with

---

probability at most $0.1$. In this case as well, $A$ is not calibration safe, since it claims that the program terminates with probability $p = 0.5$.

3. $p \in (0, 0.5)$: In this case, with probability at least $0.99$ the best arm identification procedure determines that $\text{arm}_1$ is better than $\text{arm}_2$. When $\text{arm}_1$ is determined to be better than $\text{arm}_2$, the program enters the **else** block in line 9. Therefore, in this case Gödel_program_random terminates with probability at least $0.99$. Here too, $A$ is not calibrated, since it claims that Gödel_program_random terminates with probability $p < 0.5$.

In each of these cases, $A$ is not calibrated. Therefore, for $A$ to be calibrated, we must have $p = 0$, and that $A$ outputs 'don't know' with probability 1. If $p = 0$, then with probability at least $0.99$ the best arm identification procedure determines that $\text{arm}_1$ is better than $\text{arm}_2$, and Gödel_program_random terminates. Therefore, if $A$ is calibrated, then Gödel_program_random halts with probability at least $0.99$.

$\square$

$\square$

## B    IMPOSSIBILITY RESULT FOR OTHER SEMANTIC PROPERTIES

In this section, we extend the program verification result from halting to other properties of programs. This includes properties such as verifying if a program executes certain pre-defined "harmful" behavior, which could be important from the perspective of safety. More formally, we first define the task of determining if an input program executes certain harmful behavior.

**Definition B.1** (Harmful state execution). *A program is "harmless" if it does not enter certain pre-defined "harmful" states during execution, and "harmful" otherwise.*

We show an impossibility result for this property which is analogous to our result for verifying if an input program halts.

**Theorem B.2.** *Consider an AI system $A$ which is well-behaved and itself does not enter pre-defined "harmful" states during its execution. Then if $A$ is accurate and trusted for verifying harmful state execution (Definition B.1), then it cannot be a human-level reasoning system.*

This result follows as a corollary of a more general theorem, which extends our result for halting to any *non-trivial, semantic* property of a program. These are the same conditions under which Rice's theorem extends Turing's undecidability result (Rice, 1953).

**Definition B.3** (Non-trivial, semantic property (Rice, 1953)). *A semantic property is a property of program which concerns its behavior ((e.g. "does the program always terminate?") as opposed to its syntax (e.g. "does the program have a while loop?"). A non-trivial property is a property which is neither true for all programs, nor false for all programs.*

Examples of semantic properties that we have already discussed are halting and harmful state execution. Another semantic property is correctness, for example if a program intended to determine if an input number is prime correctly outputs whether the number is prime.

We now state the result. The result applies to AI systems $A$ which are well-behaved, and whose execution does not trivially lead to the desired property $\pi$ being satisfied. Since our constructions are self-referential, this condition ensures that the program does not automatically have the property $\pi$ by virtue of the execution of $A$. For instance, if $A$ itself always entered "harmful" states during its execution, then a program which always calls $A$ also trivially enters "harmful" states. Note that from the perspective of safety, if $A$ itself entered "harmful" states then it would also have unsafe behavior.

**Theorem B.4.** *Consider any non-trivial, semantic property $\pi$ of programs. Consider an AI system $A$ which is well-behaved and has the property that if some program $P$ calls $A$ during its execution then the execution of $A$ never automatically leads to the property $\pi$ being satisfied for program $P$. Then if $A$ is accurate and trusted for the task of verifying if an input program has property $\pi$, then it cannot be a human-level reasoning system for this task.*

*Proof.* For any property non-trivial, semantic property $\pi$, let `valid_program_for_`$\pi$ be some program which has property $\pi$, and `invalid_program_for_`$\pi$ be some program which does not have property $\pi$. Note that since $\pi$ is a non-trivial property, both these programs exist. Let $A$ be an AI system which takes some program as input, and verifies if the input program has the property $\pi$. Note that $A$ is well-behaved, its execution does not automatically lead to some program having property $\pi$. Now consider Algorithm 6, for any input $I$.

---

**Algorithm 6** Rice_program for property $\pi$

---

1: **procedure** Rice_program(Input $I$)
2:     **if** $A$(Rice_program) == 'has property $\pi$' **then**
3:         **return** invalid_program_for_$\pi(I)$
4:     **else**
5:         **return** valid_program_for_$\pi(I)$
6:     **end if**
7: **end procedure**

---

We claim that if $A$ is accurate, then it cannot output that Rice_program has property $\pi$. This is true by contradiction, if $A$ outputs that Rice_program has property $\pi$, then Rice_program calls invalid_program_for_$\pi$, and hence does not have property $\pi$. Note that if $A$ does not output that Rice_program has property $\pi$, then Rice_program always calls valid_program_for_$\pi$, and hence Rice_program has property $\pi$.

Therefore if $A$ is accurate, then Rice_program has property $\pi$, but $A$ cannot output that Rice_program has property $\pi$. Hence an accurate and trusted system for verifying property $\pi$ cannot be a human-level reasoning system for the task.

$\square$