# OpenReview forum: "Limitations on Safe, Trusted, Artificial General Intelligence"
_ICLR.cc/2026/Conference — Submitted to ICLR 2026_

### Official Review · Reviewer_8cN4 · 2025-10-21

**Soundness:** 3
**Presentation:** 3
**Contribution:** 3
**Rating:** 6
**Confidence:** 2

**Summary:**

The paper gives strict, mathematical definitions of safety (“never makes a false claim”), trust (“we assume the system is safe”), and AGI (“solves every instance a human can provably solve with non-zero probability”). With these definitions, it reaches the conclusion that an AI system that is both safe and trusted cannot be an AGI. In other words, among {safety, trust, AGI}, any system can have at most two. The proofs adapt Gödel-style diagonalization and Turing’s halting arguments to concrete AI tasks: program verification, planning, and graph reachability, constructing instances humans can certify but any safe, trusted system must abstain on.  They also analyze a relaxed notion, calibration-safety, and prove a parallel impossibility: even well-behaved, calibration-safe, trusted systems must abstain on some provably halting cases.  Discussion emphasizes the worst-case nature and that “adding axioms” only pushes the problem one step further due to self-reference.

**Strengths:**

1. Very clear.  It gives crisp, operational definitions of safety (“does not make any false claims… either answers… correctly or abstains”), trust (“the assumption that the system is safe”), and AGI (“solve the instance with some non-zero probability”), which anchors all results.

2. The paper has a clear take home message: The paper's premise is that a system cannot be safe, trusted, and AGI simultaneously. It uses clear examples: spells this out for program verification, planning, and graph reachability.

3. Discusses other approaches with relaxed safety. It introduces calibration-safety and still derives an impossibility (Theorem 4.2): even a well-behaved, calibration-safe, calibration-trusted system must abstain on some instances that provably halt with high probability.

4. The proofs are presented in a way that doesn't require deep background in formal logic or axiomatic systems, making them accessible to the broader AI research community while maintaining mathematical rigor.

**Weaknesses:**

Artificial Trust Assumption: The "trust" assumption (that we scientifically accept the system is safe) is somewhat circular in the construction. In practice, we can empirically test AI systems on specific instances without assuming global safety, which could allow us to solve the constructed adversarial examples.

Questionable AGI Definition: The definition of AGI (matching humans on all provably solvable instances) is narrow and controversial. It doesn't capture common intuitions about AGI involving general reasoning, learning, and adaptation across diverse domains. Many would argue a system could be "AGI" while failing on specific adversarial instances.

Overly Restrictive Definitions: The definition of safety (zero false claims, must be correct or abstain) is extremely stringent and may not reflect how "safe" AI systems are conceptualized in practice. Real-world safety often involves bounded error rates, not perfect correctness.

Limited practical impact: The proofs rely on self-referential constructions; the paper does not estimate their prevalence in real deployments or offer empirical benchmarks, leaving the practical incidence of the impossibility unmeasured.

**Questions:**

Can you characterize trade-offs (e.g., bounds on abstention rate vs. safety) rather than all-or-nothing impossibility?

Which weaker safety/trust notions (e.g., probabilistic safety, local trust, distributional assumptions) still yield impossibility—and which permit meaningful AGI capability?

Any empirical proxies (benchmarks) that approximate your constructions to test how real systems fail/abstain?

---

> ### Author Response · Authors · 2025-11-20
>
> We thank the reviewer for their comments and feedback. Below we discuss the major concerns. We have consolidated responses to some remarks which appeared both under “weaknesses” and “questions”.
>
> *Regarding definitions being restrictive*
>
> We have decided to substitute “safety” with “accuracy”, and “AGI” with “human-level reasoning”. This avoids unnecessary  conflation with broad and popular terms such as “safety” and “AGI”. Though “accuracy” loses some of the semantics we had in mind with “safety”, we have added some discussion to maintain that connection, and similarly with the substitution of “AGI” with “human-level reasoning”.
>
> *Regarding limited practical impact*
>
> We believe that before achieving guarantees such as safety, trust, and capability in real-world AI systems, it is important to first understand the limits of what is possible. There is already a lot of important and influential work being done in the field to achieve safety, trust and capabilities in practice, as we also discuss in our related work section. This work is extremely crucial, but our goal here is very different, which is to outline the boundaries of possibilities. Because of the scale of impact that AI can have, understanding its limits is fundamental and important ---  in the same way that understanding the limits of mathematics or computation is important (as was done in Godel’s and Turing’s work, which this paper is itself based on).
>
> *Which weaker safety/trust notions (e.g., probabilistic safety, local trust, distributional assumptions) still yield impossibility—and which permit meaningful AGI capability?*
>
> We consider one such relaxation, calibration, in Section 4. Calibration only requires the system to very approximately (up to +/- 0.25 error) determine the probability of a program halting. We show that even with this relaxation, we get an impossibility result (Thm 4.2). It would be an interesting direction for future work to consider other relaxations, and see if those also yield some impossibilities.
>
> *Empirical proxies and trade-offs between notions*
>
> This would be a very interesting direction for future work. It is an important question, but one which seems inherently domain and problem-dependent, and requires domain-expertise to answer. For instance, for an LLM used as a character in a game there is a much higher tolerance for error, than for a LLM used to give medical advice. Investigating meaningful domain-dependent safety notions would be interesting, but is distinct from our goal here, which is to outline the broad limits of what is possible and not possible.
>
> We thank the reviewer again for their feedback and engagement with the paper. If there are any questions or concerns which are outstanding, we would be very happy to discuss further.

---

### Official Review · Reviewer_FUx6 · 2025-11-01

**Soundness:** 2
**Presentation:** 3
**Contribution:** 2
**Rating:** 4
**Confidence:** 3

**Summary:**

This paper proposes a formal impossibility result showing that safety, trust, and Artificial General Intelligence (AGI) cannot be simultaneously achieved. The authors formalize safety as the system never making false claims, trust as the assumption that the system is safe, and AGI as always matching or exceeding human capability across all provably solvable task instances. Using Gödel’s incompleteness theorem and Turing’s halting problem as theoretical foundations, they prove that any AI system that is both safe and trusted cannot be an AGI, illustrating this through program verification, planning, and graph reachability tasks. The paper aims to reveal a fundamental trade-off between correctness guarantees and generality.

**Strengths:**

- The paper attempts a rigorous formal treatment of concepts - safety, trust, and AGI - that are often discussed only qualitatively.

- The argument is structured across multiple canonical AI problem settings, enhancing clarity and accessibility.

- The connection to Gödel’s and Turing’s classical results is elegant and provides theoretical depth.

**Weaknesses:**

- The definitions of safety and trust are overly strong and conceptually narrow, assuming perfect correctness and universal acceptance of safety: conditions far removed from realistic AI deployments.

- The reduction of trust to “assumption of safety” is philosophically and technically weak, ignoring the social, epistemic, and empirical bases of how trust in AI is actually formed.

- The core conclusion depends entirely on these rigid definitions, so the incompatibility may not hold under more plausible, probabilistic, or bounded interpretations of safety and trust.

- The paper implicitly equates truth with safety, yet not all true claims are safe (e.g., truthful but uncalibrated or context-insensitive outputs can still lead to unsafe outcomes). This undermines the claim that correctness alone guarantees safety.

- While the logical result is internally consistent, the external validity and practical relevance of the theorem are insufficiently supported.

**Questions:**

- How would the impossibility result change if safety were defined probabilistically or contextually (e.g., bounded-risk safety instead of absolute correctness)?

- Given that not all true statements are safe, could the authors refine the framework to distinguish epistemic correctness from operational safety?

- Would redefining trust as empirical confidence (based on verification or transparency) rather than assumption preserve or dissolve the incompatibility?

- Could these results extend to learning-based systems where reasoning is inductive and not explicitly logical, and if so, how would "provable correctness" be replaced?

---

> ### Author Response · Authors · 2025-11-20
>
> We thank the reviewer for their comments and feedback. Below we discuss the major concerns. We have consolidated responses to some remarks which appeared both under “weaknesses” and “questions”.
>
> *Regarding definitions being narrow*
>
> We have decided to substitute “safety” with “accuracy”, and “AGI” with “human-level reasoning”. This avoids unnecessary  conflation with broad and popular terms such as “safety” and “AGI”.Though “accuracy” loses some of the semantics we had in mind with “safety”, we have added some discussion to maintain that connection, and similarly with the substitution of “AGI” with “human-level reasoning”.
>
> *Regarding practical relevance of the theorem*
>
> We believe that before achieving guarantees such as safety, trust, and capability in real-world AI systems, it is important to first understand the limits of what is possible. There is already a lot of important and influential work being done in the field to achieve safety, trust and capabilities in practice, as we also discuss in our related work section. This work is extremely crucial, but our goal here is very different, which is to outline the boundaries of possibilities. Because of the scale of impact that AI can have, understanding its limits is fundamental and important ---  in the same way that understanding the limits of mathematics or computation is important (as was done in Godel’s and Turing’s work, which this paper is itself based on).
>
> *How would the impossibility result change if safety were defined probabilistically or contextually (e.g., bounded-risk safety instead of absolute correctness)?*
>
> We consider one such relaxation, calibration, in Section 4. Calibration only requires the system to very approximately (up to +/- 0.25 error) determine the probability of a program halting. We show that even with this relaxation, we get an impossibility result (Thm 4.2). It would be an interesting direction for future work to consider other relaxations, and see if those also yield some impossibilities.
>
> *Given that not all true statements are safe, could the authors refine the framework to distinguish epistemic correctness from operational safety?*
>
> Interesting question, and it has led to some new results in our paper! First, we note that our goal is to establish a necessary condition for safety. We recognize that safety can have many facets, and our condition need not be sufficient. Note that since we show an impossibility result, imposing more conditions on a system in order for it to be safe still retains the same impossibility.
>
> That being said, to consider a more operational version of safety, we considered program verification tasks beyond just halting. Specifically, in the new Appendix B, we defined the task of determining “Harmful state execution” (Def B.1). This is a more operational notion as the reviewer had suggested. The goal here is to determine if an input program enters certain pre-defined “harmful” states in its execution. The reason for this being more operational is that these harmful states could be defined with specific operational notions of safety in mind. Theorem B.2 shows an impossibility for this notion. This result itself is a corollary of a new result, Thm B.4, which generalizes our result for halting to any “semantic” property of a program.
>
> We thank the reviewer for raising this concern, and hope the result helps (at least partly) address the concern.
>
> *Would redefining trust as empirical confidence (based on verification or transparency) rather than assumption preserve or dissolve the incompatibility?*
>
> Our results require that there is an assumption of safety. This assumption could itself arise from empirical investigations. As long as there is an assumption of safety (perhaps from empirical investigations), the incompatibility is still present.
>
> *Could these results extend to learning-based systems where reasoning is inductive and not explicitly logical, and if so, how would "provable correctness" be replaced?*
>
> Our result is agnostic to whether the system is inductive or logical, or even to whether it is possible or not to show provable correctness for the system. As discussed above, trust could arise only from empirical investigations of the system which may be true for learning-based systems, in which case the incompatibility still persists.
>
> We thank the reviewer again for their feedback and engagement with the paper. If there are any questions or concerns which are outstanding, we would be very happy to discuss further.

---

### Official Review · Reviewer_yQV8 · 2025-11-02

**Soundness:** 2
**Presentation:** 3
**Contribution:** 2
**Rating:** 2
**Confidence:** 4

**Summary:**

This paper presents some impossibility results on concrete definitions of safe, trusted, and general artificial intelligence (AI).

An AI system is formalized as a program that receives some input and produces an output that can be "yes", "no", or "don't know". An AI system is safe if its outputs are always correct or "don't know", trusted if it is generally assumed to be safe (by whom?), and general if it can solve any task that a human can solve.

The paper shows several such impossibility results, following classical arguments to show the undecidability of fundamental problems such as the halting problem.

**Strengths:**

**S1.** The results and proofs are sound.

**S2.** Once you buy the premise of the language being used, the paper is easy to follow.

**S3.** The paper provides a fun perspective on the theoretical limits of computer programs compared to human reasoning, that will be appealing for researchers acquainted with classical undecidability arguments, and may bring attention to them to unfamiliar readers in the AGI space.

**Weaknesses:**

**W1. Eye-catching language, that overpromises and underdelivers in the results**.
The language of safe, trust, and AGI is not a good choice. These concepts are subject to a wide study, and this paper does not engage properly with them.

In particular, sentences like "an AI system that is safe and trusted cannot be an AGI" are very eye-catching; however, when you properly understand them, they are not surprising at all.

**W2. The treatment of "trust" is not very solid.**
The notion of "trust" is treated informally and conflated with scientific acceptability, which has well-understood epistemic and community-dependent aspects. The authors should either formalize trust as a belief layer and avoid claims that invoke vague phrases such as "scientifically accepted proof." A better approach would be to treat "trust" as a belief layer, so it is different to say "A is safe", than "A is believed to be safe". This also leads to different concepts of AGI: a program that can solve any problem a human can, or a problem that can solve any problem that a human believes they can solve.

*The problem with the "science" talk*. Expressions like "scientifically accepted that the system is safe", or "scientifically acceptable proof" are meaningless in essence. To go a step deeper, in most contexts a scientific truth or a scientifically acceptable statement is understood to be a statement that is supported by the available evidence and part of the consensus of the scientific community for which it is relevant. There are two key notions to treat here: data and community. Data is very important because these statements typically refer to empirical sciences, where absolute (or formal) truths cannot exist and are always subject to fallibility: for any scientifically accepted statement, there exists an experiment and an adequate result that turn the statement to unaccepted. Computer science (and more generally, mathematics) does not depend on data because the statements are true or false independently of the data. Community is also important because a scientifically acceptable statement is always meant in reference to a community. The same fact may be acceptable for a community and unacceptable for another one, so saying that a fact is scientifically accepted is meaningless without mentioning to which community it refers.

Because of all these arguments, I would strongly recommend the authors to explicitly include a formal way to reason about beliefs in their paper, instead of relying on this vague and imprecise language of something being "scientifically acceptable".

**W3. Modelling of human reasoning**. By definition (the definition adopted in this paper, not necessarily the generally accepted definition), the human's problem-solving capability subsumes that of the AI (since the human can incorporate the AI as a tool). Thus, a claim like "humans can always solve at least as much as a safe, trusted AI" would be tautological. This is not clear from an initial reading of the paper, and it would be helpful if the authors explicitly discussed this asymmetry and its implications for the notion of AGI they adopt.

**W4. Technical novelty**. The results are reformulations of classical diagonalization arguments known for years in the computability literature. The novelty is therefore primarily rhetorical rather than technical.

**Questions:**

**Q1**. In Algorithm 1, it is never defined what a valid input to a program is, and whether it is something that is reasonable to check. Would it be possible that checking whether an input is valid is itself an undecidable problem, and therefore $\mathtt{Goedel-program}$ is not well defined?

**Q2**. Since the graph reachability problem cannot be solved in general faster than in the time it takes to read the whole graph, any program that solves it faster and in a "safe" way is forced to leave some instances "unanswered". A human could arbitrarily select one of these instances, add it to its solving capabilities in constant time, and thus claim to be able to solve more instances than the AI program in a given time, concluding then that it cannot be an AGI. Is this a fair restatement of theorem 3.10?

**Details Of Ethics Concerns:**

This paper can make very bold statements about "safe" and "trusted" AI because it is using very narrow definitions of terms that are used in the field to describe broad requirements with societal impact. I'm not sure if this is supposed to be considered an ethics concern, but publishing papers with this language on top-tier research venues will be problematic for the research community on safe and trustworthy AI.

---

> ### Author Response · Authors · 2025-11-20
>
> We thank the reviewer for their comments and feedback. We discuss some comments further below.
>
> *(W1) Eye-catching language, that overpromises and underdelivers in the results.*
>
> Thank you for raising this criticism, we have modified the paper to address them. In the original submission, we tried to be very upfront in the abstract itself that our results apply to strict formal interpretations of notions such as safety and AGI, since these terms have a very wide scope associated with them. Since the reviewer feels that the language was still eye-catching, we have decided to substitute “safety” with “accuracy”, and “AGI” with “human-level reasoning”. Though “accuracy” loses some of the semantics we had in mind with “safety”, we have added some discussion to maintain that connection, and similarly with the substitution of “AGI” with “human-level reasoning”. With these changes, we hope the reviewer feels that we are promising closer to what we are delivering.
>
> *(W2) The treatment of "trust" is not very solid.*
>
> Thanks for the comment. By trust, we are referring to the assumption that the system is safe. We agree that some of the mentions of “scientific acceptability” are unnecessary and confusing, and have revised them. In the proof of Thm 3.2 we have clarified that for a safe, trusted system a human can prove that Godel_program is well-behaved. The discussion of scientific acceptability is unnecessary here, as the reviewer remarked.
>
> As a slight digression, we prefer to regard trust as an assumption, since even in mathematics (or computer science) all results are still based on axioms (such as Peano Arithmetic or ZFC) which we assume to be consistent, even though they cannot be proven to be consistent (due to Godel’s results again). We have no proof that mathematics or arithmetic is consistent and does not lead to contradictions, but mathematics proceeds by assuming consistency (in case the reviewer or any future reader is interested in reading further about this, this post about consistency of mathematics is quite nicely written https://www.mathpages.com/home/kmath347/kmath347.htm )
>
> *(W3) Modelling of human reasoning.*
>
> Good point. Our goal here is to show a separation that there are certain task instances solvable by humans but cannot be solved by a safe, trusted AI. From this perspective, task instances which the safe, trusted AI can solve are not of as much interest to us. But the reviewer is correct that if we consider the human to be a human+AI system (where the AI system is safe and trusted), then that combination is always at least as good as just the safe, trusted AI system. We can add this discussion to the paper.
>
> *(W4) Novelty is therefore primarily rhetorical rather than technical.*
>
> We believe that our main novelty is in the definitions, not necessarily in the rhetoric. We also note that while we are obviously standing on the shoulders of giants by using Godel’s and Turing’s results, some technical care is needed to use the diagonalization arguments, such as in the calibration definition (Thm 4.2) which requires usage of some results from multi-armed bandits.
>
> *(Q1) Would it be possible that checking whether an input is valid is itself an undecidable problem, and therefore  is not well defined?*
>
> This only requires a type check, for example does program P take as input two arguments or is it only defined to take a single one input. More formally, this is a syntactic rather than a semantic property of a program, and is (efficiently) decidable. It is possible to even skip this type check and we only kept it for clarity. If we regard the program as a Turing machine then it just takes a single input on its input tape (this is also briefly discussed below Algorithm 1). This input would just be the concatenation of P and I in Algorithm 1, and for this no explicit type check would be needed.
>
> (continued in the comment below)

---

> > ### Author Response · Authors · 2025-11-20
> > **Response to review (continued)**
> >
> > *(Q2) Interpretation and restatement of Thm 3.10*
> >
> > We thank the reviewer for raising this, since this is not a fair restatement of Thm 3.10 and helps understand the result better. The crucial aspect of our results is that they do not require humans to have access to a “secret” or “trapdoor” regarding how the instance was generated which the AI system does not possess. To probe further, consider the factoring task. Suppose a human generates a number $t=p*q$ where $p$ and $q$ are two large primes, and asks an AI system to factor $t$. If factoring is hard then the AI system cannot hope to solve the problem in any reasonable time, but the human who generated the instance can obviously solve the problem easily. However, note that this requires the human to have access to special information which the AI system does not possess. Crucially, note that some other human who does not know how the instance was generated cannot solve the instance. Therefore we don’t believe that such a separation between AI and humans is interesting.
> >
> > In all our results, when we say the task is solvable by humans, we mean that there is a proof of the solution, which any (mathematically-proficient) human can write in principle. In particular, the proofs only require that the system is safe and trusted, and no other special information or secret about how the instance was generated or selected.
> >
> > The reviewer's interpretation of Thm 3.10 is similar to the above factoring example, in the sense that one human can construct this task instance which the AI system cannot solve in time less than $T$, but there is no reason to believe that this task instance is more universally solvable by humans in constant time. However, what we show is stronger --- for a safe, trusted system we construct an explicit instance for which a human can write a provable solution without any special knowledge of how the instance was generated. There is universality in the human’s proof, and other humans can come to the same conclusion if the system is safe and trusted.
> >
> > We thank the reviewer again for their feedback and engagement with the paper. If there are any questions or concerns which are outstanding, we would be very happy to discuss further.

---

> > ### Comment · Reviewer_yQV8 · 2025-11-26
> >
> > I thank the authors for their thorough response.
> >
> > I thank them for not using the terms "safe" and "AGI" in the revised manuscript. I agree that "accurate" is also not very satisfying, and I would suggest to use "sound" instead. In the world of logic and formal verification, an algorithm is sound when it produces correct answers, although it may sometimes not produce an answer at all. This sounds exactly like the notion of "safe" that you are using.
> >
> > I appreciate the response to Q1, and the acknowledgement of W3. With respect to W4, I may have been too harsh with respect to the technical novelty.
> >
> > With respect to W2, I see your point that mathematics does have results based on data (as in, the data suggests that ZFC is consistent, but the data may contradict that in the future). I guess if you are going for such strong assumptions, there is some consolidated pieces of software that are considered "trusted" nowadays, but I don't see how any general purpose AI would ever be trusted, at least not in our lifetime. In any case, ss far as I can see, this notion of "trust" is also not precise, and it is also not aligned with the use of the term in the literature.
> >
> > If your notion of trust is akin to a mathematical assumption, shouldn't the theorems be written as "if A is sound, and under the assumption that A is sound, then A is not AGI", i.e., "A sound implies A is not AGI". What am I missing?

---

> ### Author Response · Authors · 2025-11-27
>
> We thank the reviewer for their engagement.
>
> *Soundness as opposed to accuracy*
>
> We did consider soundness as an alternative to safety, instead of accuracy. We felt though that soundness gives the impression of internal consistency (such as for an axiom system), whereas our notion only asks the AI system to be correct on the given instance if it does not abstain. Since our results do not concern only formal reasoning or axiom-based systems, soundness seems slightly out of place to us. But we will continue thinking about it, and welcome further thoughts that the reviewer may have on it and are open to suggestions.
>
> *If your notion of trust is akin to a mathematical assumption, shouldn't the theorems be written as "if A is accurate, and under the assumption that A is accurate, then A is not AGI", i.e., "A sound implies A is not AGI".* (we edited the comment to use accurate instead of sound, to be consistent with our current notation)
>
> What we show are the following 2 claims (this is in the context of the first proof, Thm 3.2, but holds across other proofs):
>
> Claim 1: If A is accurate, then Godel_program is well behaved, but A cannot output that Godel_program is well behaved.
>
> Claim 2: If A is accurate and trusted by human(s), then human(s) can prove that Godel_program is well behaved.
>
> For clarity, we can restate Claim 2 without using "trust" in case it helps.
>
> Claim 2' (equivalent to Claim 2): If A is accurate and assumed to be accurate by human(s), then human(s) can prove that Godel_program is well behaved.
>
> Claim 1 and Claim 2/2' together imply that if A is accurate and trusted, then it is not AGI.
>
> Note that the following claim (which is possibly suggested by the reviewer's comment) is not necessarily true.
>
> Claim 3: If A is accurate, then human(s) can prove that Godel_program is well behaved.
>
> The reason Claim 3 is not true is that accuracy is only an intrinsic property of system A. To make inferences about Godel_program, it is necessary to also have the assumption that A is accurate (which is a statement about A's epistemic status). If A is in reality accurate but it is not assumed to be accurate, then human(s) cannot necessarily come to the conclusion that Godel_program is well behaved.
>
> Thank you once again, and please let us know if you have any further questions or concerns.

---

### Official Review · Reviewer_AuXo · 2025-11-11

**Soundness:** 3
**Presentation:** 4
**Contribution:** 2
**Rating:** 6
**Confidence:** 3

**Summary:**

This paper formally defines safety (no false claims, either correct answers or abstention), trust (assumption of safety), and AGI (matching or exceeding human capability on all provably solvable task instances) for AI systems. Its core finding is the fundamental incompatibility of these three properties: a safe and trusted AI system cannot be an AGI system, as there exist task instances (in program verification, planning, and graph reachability) easily solvable by humans but unsolvable by the system. The proofs draw parallels to Gödel’s incompleteness theorems and Turing’s halting problem undecidability, and the result extends to a relaxed "calibration-safety" notion. The paper also discusses potential workarounds (e.g., adding axioms) and clarifies that the result is worst-case rather than universal.

**Strengths:**

+ The paper provides precise mathematical definitions for vague core concepts (safety, trust, AGI), addressing a longstanding gap in AGI and AI safety research.

+ By linking classic computability results (Gödel, Turing) to modern AI challenges, it reveals a fundamental constraint on AGI development that is independent of specific technologies or architectures.

+ The result is validated across three critical AI tasks (program verification, planning, graph reachability), enhancing its generalizability and practical relevance.

+ Exploring calibration-safety (a relaxed safety notion) shows the incompatibility is not limited to strict safety, strengthening the core argument.

**Weaknesses:**

- The AGI definition focuses on "matching human capability on provably solvable instances", which excludes aspects like adaptability, common sense, or learning from unstructured data—key parts of mainstream AGI discourse.
- The strict mathematical assumptions (e.g., absolute safety, perfect trust) may not align with practical AI deployments, where "sufficiently safe" or "empirically trusted" systems are more common. Note that even the traditional software system could have bugs or other trustworthiness issues. (However, I understand that the author aims to present their argument from a universal and fully safe perspective.)
- The paper assumes humans can consistently produce "scientifically acceptable proofs" for task instances, which I think might be a little bit overlooking human cognitive limitations (e.g., bias, resource constraints) highlighted in cognitive science.
- While identifying the incompatibility, the paper offers little guidance on tradeoffs or implementation strategies for balancing safety, trust, and capability in real-world AI systems.

**Questions:**

1. How might the paper’s safety-trust-AGI incompatibility shift if AGI prioritizes "most" (not "all") human-solvable tasks (matching real-world needs)? If so, which task subsets could work for safe/trusted setups? Do the authors share insights on this?

2. Could hybrid systems (e.g., AI augmented with human oversight for edge cases) circumvent the incompatibility, and what formal guarantees might apply to such configurations?

3. The paper notes relaxed safety/trust interpretations could enable partial compatibility—what specific thresholds for "acceptable error" or "empirical trust" would make AGI feasible while retaining meaningful safety?

4. How do the results extend to modern foundation models (e.g., LLMs) that often abstain implicitly (via vague outputs) rather than explicitly saying "don’t know", and how would calibration-safety apply to their probabilistic outputs?

5. Given the self-referential nature of the constructed task instances, are these edge cases relevant to practical AGI applications, or do they remain theoretical constraints with limited real-world impact?

---

> ### Author Response · Authors · 2025-11-20
> **Response to review**
>
> We thank the reviewer for their comments and feedback. Below we discuss the major concerns. We have consolidated responses to some remarks which appeared both under “weaknesses” and “questions”. We would also like to point the reviewer to the general response, which addresses some questions around our notions too.
>
> *While identifying the incompatibility, the paper offers little guidance on tradeoffs or implementation strategies for balancing safety, trust, and capability in real-world AI systems*
>
> We believe that before achieving guarantees such as safety, trust, and capability in real-world AI systems, it is important to first understand the limits of what is possible. There is already a lot of important and influential work being done in the field to achieve safety, trust and capabilities in practice, as we also discuss in our related work section. This work is extremely crucial, but our goal here is very different, which is to outline the boundaries of possibilities. Because of the scale of impact that AI can have, understanding its limits is fundamental and important, in the same way that understanding the limits of mathematics or computation is important (as was done in Godel’s and Turing’s work, which this paper is itself based on).
>
> *Given the self-referential nature of the constructed task instances, are these edge cases relevant to practical AGI applications, or do they remain theoretical constraints with limited real-world impact?*
>
> Good question, as we also discuss in the 2nd point in the discussion (Section 4), self-referential calls to the AI system may not be implausible when systems have general-purpose capabilities and can be used to solve multiple tasks. Also as argued in the previous point and in the general response, we believe that it is important to outline the limits of what is possible with AI systems. We note that Godel’s and Turing’s constructions are also self-referential, but are still important barriers for mathematics and computation.
>
> *How might the paper’s safety-trust-AGI incompatibility shift if AGI prioritizes "most" (not "all") human-solvable tasks (matching real-world needs)? If so, which task subsets could work for safe/trusted setups? Do the authors share insights on this?*
>
> Thanks for the interesting question. Our impossibility result for graph reachability arose from an attempt to consider simpler tasks to see if there is still a separation for them. Determining halting is an undecidable problem but graph reachability can be solved in polynomial time, and hence is much lower in the computability/complexity hierarchy. However, Thm 3.10 shows that even for such tasks there are instances such that the safe, trusted AI system needs the trivial amount of time to solve the problem, whereas the problem can be provably solved by humans with a short proof. An interesting direction of future work is to consider tasks or problems at other levels of complexity.
>
> *Could hybrid systems (e.g., AI augmented with human oversight for edge cases) circumvent the incompatibility, and what formal guarantees might apply to such configurations?*
>
> It would be interesting to formalize human-AI systems and understand their limits. In some sense, a separation like we show may not be possible for a hybrid human-AI system --- since such a system could always solve a problem which is solvable by humans, by reverting to the human for an answer.
>
> *The paper notes relaxed safety/trust interpretations could enable partial compatibility—what specific thresholds for "acceptable error" or "empirical trust" would make AGI feasible while retaining meaningful safety?*
>
> This is an important question, but one which seems inherently domain and problem-dependent, and requires domain-expertise to answer. For instance, for an LLM used as a character in a game there is a much higher tolerance for error, than for a LLM used to give medical advice. Investigating meaningful domain-dependent safety notions would be interesting, but is distinct from our goal here, which is to find fundamental limits on what is possible.
>
> (continued in the comment below)

---

> > ### Author Response · Authors · 2025-11-20
> > **Response to review (continued)**
> >
> > *How do the results extend to modern foundation models (e.g., LLMs) that often abstain implicitly (via vague outputs) rather than explicitly saying "don’t know", and how would calibration-safety apply to their probabilistic outputs?*
> >
> > Our results extend to any semantically equivalent way of saying “don’t know” or giving an answer. This is because the if condition in our construction can be replaced by a check for semantic equivalence to the answer if the LLM does not conclude its response with a strict  “don’t know”, for example one could map all vague outputs to “don’t know”. Similarly, if the LLM expresses its confidence in the answer either numerically or in words (such as “fully sure”, “somewhat sure” etc.), the impossibility result for calibration-safety still applies since it rules out even very coarse calibration (outputting $p \pm 0.25$, when true probability is $p$).
> >
> > We thank the reviewer again for their feedback and engagement with the paper. If there are any questions or concerns which are outstanding, we would be very happy to discuss further.

---

### Author Response · Authors · 2025-11-20
**Summary of changes**

We thank all the reviewers for their comments and feedback. We have responded individually to all reviewers, and are summarizing some of the changes in the paper below.

*Changes to the choice of “safety” and “AGI”*

The reviewers had raised issues with our choice of terms, especially “safety” and “AGI”. We have decided to substitute “safety” with “accuracy”, and “AGI” with “human-level reasoning”. This avoids unnecessary  conflation with broad and popular terms such as “safety” and “AGI”. Though “accuracy” loses some of the semantics we had in mind with “safety”, we have added some discussion to maintain that connection, and similarly with the substitution of “AGI” with “human-level reasoning”.

*Additional result on more operational implications for safety*

Reviewer FUx6 had asked us to consider a more operational version of safety. We have added some new results for this, and considered program verification tasks beyond just halting. Specifically, in the new Appendix B, we defined the task of determining “Harmful state execution” (Def B.1). This is a more operational notion as the reviewer had suggested. The goal here is to determine if an input program enters certain pre-defined “harmful” states in its execution. The reason for this being more operational is that harmful states could be defined with specific operational notions of safety in mind. Theorem B.2 shows an impossibility for this notion. This result itself is a corollary of a new result, Thm B.4, which generalizes our result for halting to any “semantic” property of a program.

*Implications of the results*

We have added more discussion on implications of the results in Section 4. To repeat some of that discussion, by way of analogy, Gödel's and Turing's results pointed to fundamental barriers to mathematics and computation. Though these barriers were worst-case, they identified the limits of what is possible and what is possible.  Similarly, we believe that it is important to outline the fundamental limits of what is possible in the context of AI and requirements of accuracy and trust. Standing on the shoulders of giants (Godel and Turing), already brings to view fundamental barriers with regards to this, and this is what our paper presents. Given the significant interest and economic capital being devoted to building accurate, safe or reliable human-level reasoning, we believe it is important to understand the barriers fundamental to any such technology.

We thank all the reviewers again for their feedback and engagement with the paper. If there are any questions or concerns which are outstanding, we would be very happy to discuss further.

---

### Meta-Review · Area_Chair_sy8r · 2026-01-07

**Summary:**

Concerns about definitions and terminology: Multiple reviewers noted that the definitions of safety/trust/AGI are extremely narrow and misaligned with how these concepts are commonly used in the field. As a result, the central claim that a safe and trusted AI cannot be AGI was viewed as unsurprising or close to definitional once the assumptions are fully understood.

Weak formalization of trust: The treatment of trust as merely an assumption of safety was criticized as insufficient, potentially circular, and lacking a clear epistemic or social formalization, with several reviewers finding this aspect underdeveloped.

Limited technical novelty: Reviewers pointed out that the core results largely rephrase classical diagonalization arguments from Gödel and Turing, with limited new technical insight; the main contribution was seen as rhetorical or definitional rather than methodological.

Lack of practical relevance and empirical grounding: The assumptions of perfect correctness and global trust were considered unrealistic for real-world AI systems, and the self-referential constructions were viewed as having unclear relevance to practical deployments.

**Reviewer Concerns:**

Concerns about definitions and terminology: The rebuttal partially addressed this by changing terminology and acknowledging reviewer concerns. However, the underlying definitions remain narrow, and the concern that the main claim is close to definitional remains outstanding.

Weak formalization of trust: The rebuttal clarified trust as an epistemic assumption and removed confusing language, which addressed some reviewer criticisms. Still, the lack of a richer or non-circular formalization of trust remains unresolved.

Limited technical novelty: The authors argued that novelty lies in the definitions and extensions, but concerns that the results largely restate classical arguments remain largely outstanding.

Lack of practical relevance and empirical grounding: The rebuttal acknowledged this limitation and framed the work as worst-case theory, but did not substantively address concerns about realism or practical relevance, which remain outstanding.

**Reviewer Scores:**

Overall, the rebuttal would likely not change reviewer scores substantially: more positive reviewers would maintain their scores, borderline reviewers might gain modest confidence, and the most critical reviewer would likely remain unconvinced, as the core concerns about definitions, novelty, and practical relevance were not fundamentally resolved.

---

### Decision · Program_Chairs · 2026-01-26

Reject